

# Combining lime and organic amendments based on titratable alkalinity for efficient amelioration of acidic soils

Birhanu Iticha[1], Luke M. Mosley[2], Petra Marschner[1]

[1]School of Agriculture Food and Wine, The University of Adelaide, SA 5064, Australia

[2]Acid Sulfate Soils Centre, School of Biological Sciences, The University of Adelaide, SA 5064, Australia

*Correspondence to*: Birhanu Iticha (birhanuiticha.ayanssa@adelaide.edu.au)

**Abstract.** Ameliorating soil acidity using a combination of lime and organic amendments (OAs) can be an alternative to lime alone, but determining the appropriate OA rates can be difficult. We developed a new method for calculating the combined application rate of lime and OAs (wheat straw, faba bean straw, blended poultry litter, biochar, and compost),

based on the titratable alkalinity of OA and the equilibrium lime buffer capacity ($LBC_{eq}$) of acidic soils. The effect of calculated soil amendment rates on soil pH was validated at soil water contents of 60, 100, and 150% field capacity (FC). The soil used to develop and validate the method was a sandy loam with $pH_W$ 4.84 and $pH_{Ca}$ 4.21. The $LBC_{eq}$ of the soil was 1657 mg $CaCO_3$ $kg^{-1}$ $pH^{-1}$. The titratable alkalinity of the OAs ranged from 11.7 cmol $H^+_{eq}$ $kg^{-1}$ for wheat straw to 357 cmol $H^+_{eq}$ $kg^{-1}$ for compost. At 60% FC, faba bean and wheat straw amendment increased soil $pH_W$ to 6.48 and 6.42, respectively,

whereas less biodegradable or resistant OAs (ROAs) (i.e. blended poultry litter, biochar, and compost) had lower pH values. At 150% FC, the two straws increased soil $pH_W$ to only 5.93 and 5.75, respectively, possibly due to slower decomposition in submerged conditions, resulting in limited alkalinity production, whereas amendment with ROAs produced $pH_W$ values close to 6.5. With increasing lime equivalent value (LEV) of the OA, from 5.8 g $CaCO_3$ $kg^{-1}$ (wheat straw) to 179 g $CaCO_3$ $kg^{-1}$ (compost), the lime requirement to reach pHw 6.5 in lime-OA combinations decreased from 2.72 to 0.09 g $CaCO_3$ $kg^{-1}$. The

method was proved to be effective in determining appropriate rates of OAs (with or without additional lime) for management of acidic sandy loam soils in this study, but it needs to be validated for a particular soil and amendment.

**Keywords:** Lime requirement, soil amendments, alkalinity production, soil pH, lime buffer capacity, soil water content



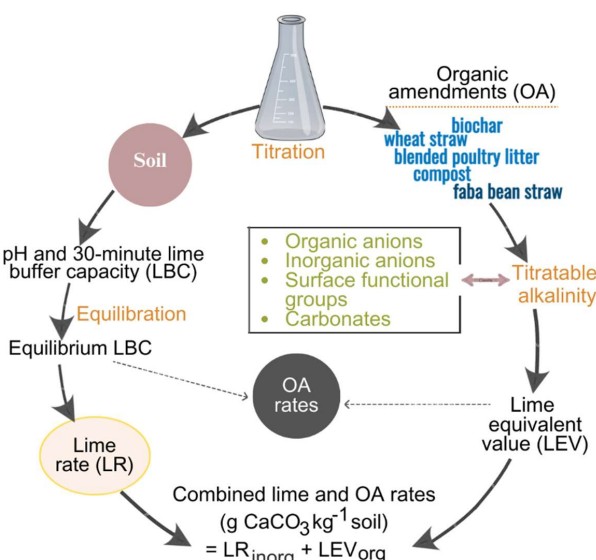

## 1 Introduction

Acidic soils with pH ≤ 5.5 are considered harmful to sensitive plants. Soil acidification can be caused by various biogeochemical processes such as oxidation and acid-dissolution reactions, root exudates, solubilisation and hydrolysis of $Al^{3+}$ which releases $H^+$, and leaching of cations ($K^+$, $Ca^{2+}$, $Na^+$, $Mg^{2+}$) (Brady and Weil, 2016; Goulding, 2016; Marschner and Noble, 2000; Mosley et al., 2014). Soil acidification in agricultural systems can also be attributed to high inputs of urea and ammonium ($NH_4^+$)-containing fertilisers where protons ($H^+$) are generated during nitrification of $NH_4^+$ to nitrate ($NO_3^-$)

and by $NH_4^+$ uptake by plants as well as by removal of alkalinity by removing plant material at harvest (Bolan et al., 2003; Hume et al., 2022; Iticha and Takele, 2019; Kunhikrishnan et al., 2016). Even a small decrease in soil pH can have a large impact on nutrient availability, nitrogen fixation by soil microbes, and sustainability of crop production (Kopittke et al., 2019).

Lime and, more recently, OAs can be used to improve the productivity of acidic soils (Garbowski et al., 2023).

Agricultural lime ($CaCO_3$) produces alkalinity ($OH^-$ and $CO_3^{2-}/HCO_3^-$ ions) and neutralises soil acidity by reacting with $Al^{3+}$ and $H^+$ ions to form $Al(OH)_3$ precipitates and $H_2O$. Organic amendments on the other hand can provide alkalinity via their surface organic functional groups (e.g. carboxylic, $-COO^-$), presence of solid and dissolved carbonates, and by release of organic and inorganic anions, which can neutralise or remove toxic ions such as $Al^{3+}$, $Fe^{3+}$, $Mn^{2+}$, and $H^+$ from the soil by reacting, complexing, and binding (Brown et al., 2008; Larney and Angers, 2012; McCauley et al., 2009). Alkalinity



production by OAs is related to the excess cation content and dissolution or decomposition rate of the amendments, which are in turn influenced by factors such as the chemical composition of the amendments and soil water content (Anderson et al., 2020; Cai et al., 2020; Védère et al., 2022).

Determination of the optimum levels of lime and OAs is necessary for efficient management of acidic soils. Lime requirement can be determined using various methods including titration, incubation, field experiments, standard buffer
solutions, and predictive equations developed from measured soil properties (Aitken and Moody, 1994; Nelson and Su, 2010). Titration is considered the most effective approach for lime recommendations (Wang et al., 2015). It involves the addition of bases such as $Ca(OH)_2$ to acidic soils in a given soil to liquid ratio (e.g. 1:5 soil to deionized water or 0.01 M $CaCl_2$ solution) followed by measurement of changes in soil pH and pH buffering capacity (pHBC) with time to predict lime requirement of the soils (Barouchas et al., 2013; Liu et al., 2004; Thompson et al., 2010). Titration results are often validated
by experiments to determine the equilibrium lime buffer capacity ($LBC_{eq}$), i.e. measure of the amount of soil acidity that must be neutralized by addition of lime to raise soil pH by one unit) in order to calculate lime requirements of acidic soils (Kissel et al., 2012).

The amounts of OAs required to neutralise soil acidity and achieve a desired pH can be calculated using the alkalinity of the amendments. Alkalinity is determined by titrating acidified OAs with a base to neutrality, using either ashed
amendments, i.e., potential or 'ash' alkalinity (Slattery et al., 1991), or air-dried amendments, i.e. titratable or available alkalinity (Feizi et al., 2017; Fidel et al., 2017; Singh et al., 2017). The ash alkalinity method overestimates alkalinity or liming potential of OAs due to the loss of anions such as sulfur and chlorine during the ashing process (Noble et al., 1996), leading to underestimation of the amounts of amendments required to neutralise soil acidity. The short-term acid neutralising effect of OAs is better determined by available alkalinity, because potential alkalinity becomes only gradually available over
a long period of time (Sakala et al., 2004).

Studies have shown that combination of lime and OAs can generate more alkalinity than lime or OA alone (Butterly et al., 2021; Lauricella et al., 2021; Li et al., 2022). However, little effort has been made to develop methods for calculating the application rates of lime combined with OAs needed to neutralise soil acidity and achieve the desired pH for plant growth. Previous research aimed at developing appropriate lime and OA combinations has been conducted in field trials, which
involve mixing different rates of lime and OAs into acidic soils and determining the response of acidity and crop yield (Celestina et al., 2018; Khoi et al., 2010; Li et al., 2019). This approach is time consuming and costly, and it is difficult to predict the soil acidity that can be neutralised. In addition, little information exists about the impact of an alkalinity-based mixture of OAs on soil pH. A laboratory method for determining lime and OAs combinations based on titratable alkalinity could shorten the time required and improve the cost-effectiveness of soil acidity amelioration.

Therefore, the first aim of this study was to determine the lime requirement of acidic soil using titration and equilibration methods as well as estimate the application rates of OAs and OAs mixes based on their titratable alkalinity. The second aim was to develop a laboratory method for calculating the lime and OA combinations required to achieve a desired soil pH, based on titratable alkalinity of the amendments and $LBC_{eq}$ of the acidic soil. Finally, to assess the effect of soil water



content on amelioration, a validation experiment was conducted by incubating acidic soil with the amendments at different

soil water contents.

## 2 Materials and Methods

### 2.1 Collection of soil and amendments

The soil used in this study was collected from the 0-10 cm layer of the non-limed treatment from the acid soil management trial site at Sandilands on the Yorke Peninsula of South Australia. It is located in the North-West of Adelaide at $34^0 33^` 14^{``}$S

latitude and $137^0 42^` 14^{``}$E longitude. The long-term mean annual rainfall and temperature of the site were 409 mm and 21.9 $^O$C. Penlime Plus™ (Angaston lime, Penrice Quarry & Mineral, South Australia) with a neutralising value of 98% was used as the lime source. The OAs used in this study were wheat straw, faba bean straw, blended poultry litter, biochar, and compost which differ in decomposability. The two straws are more decomposable than the other OAs because they haven't undergone decomposition. Due to its lower C/N ratio, faba bean straw is likely to be more readily decomposable than mature

wheat straw. The blended poultry litter comprised of equal proportions of poultry manure and sawdust/wood shavings. The biochar was made from pyrolysis (400°C) of Eucalyptus species (sourced from Green Man Char Pty Ltd, Victoria, Australia), the compost was prepared from a mixture of various organic wastes (sourced from Bunnings Pty Ltd, Australia). Blended poultry litter, biochar and compost are partially decomposed, leaving little readily available compounds for decomposition after addition to soil and are therefore referred to as poorly decomposable or resistant amendments.

The soil and OAs were dried at 30 $^0$C using a fan-forced oven. Then the oven-dried soil was crushed, sieved to pass through a 2 mm sieve size, the dried OAs were ground and sieved to 0.25-2 mm, and then used in the experiment.

### 2.2 Analysis of soil and amendments

Soil particle size distribution was determined using the hydrometer method (Bouyoucos, 1962). To measure gravimetric water content at field capacity, air-dried soil was weighed into small container with 10 cm diameter and 10 cm height,

watered to saturation, covered at the top with plastic film to prevent evaporation, and drained to a constant mass for 2 to 3 days, and reweighed. Soil pH was measured potentiometrically using a calibrated glass electrode in a soil/water (1:5) suspension ($pH_w$) and a soil/0.01 M CaCl$_2$ (1:5) suspension ($pH_{Ca}$) (Rayment and Higginson, 1992). The pH and electrical conductivity (EC) of OAs were measured in a 1:10 amendment to water ratio (Singh et al., 2017). Exchangeable acidity was determined by extracting soil with 1 M KCl solution (Pansu and Gautheyrou, 2006). The cation exchange capacity (CEC) of

the soil was determined using a colorimetric method after displacing cations with 1 M ammonium acetate (pH 7) and then extracting the ammonium ions with 1 M KCl (Carter and Gregorich, 2007). Total organic carbon (TOC) in the soil and OAs was determined using the Walkley and Black wet digestion method (Walkley and Black, 1934), whereas total N was determined using the Kjeldahl method (Bremner and Mulvaney, 1982). After digestion with concentrated nitric acid and hydrogen peroxide, the total concentrations of Fe and Al in soil as well as Ca, K, Mg, Na, P, and S in OAs were determined



105 using inductively coupled plasma optical emission spectroscopy (ICP-OES). The excess cation content in the OAs was then calculated by subtracting anions ($SO_4^{2-}$ and $H_2PO_4^-$) from cations ($Ca^{2+}$, $Mg^{2+}$, $K^+$, and $Na^+$) (Tang and Yu, 1999).

### 2.3 Collection of soil and amendments

In the Australian Soil Classification (Isbell and NCST, 2021), the soil was categorized as a Chromosol, it is classified as Lixisol in WRB (FAO, 2015). The upper A horizon which has a sandy loam texture was used in this experiment (Table 1).

110 The soil $pH_W$ and $pH_{Ca}$ measured in 1:5 soil to water/0.01 M $CaCl_2$ solution ratios are 4.84 and 4.21, respectively. The exchangeable acidity of the soil was 2.95 $cmol_c$ $kg^{-1}$. Based on the ratings of Hazelton and Murphy (2016), the soil has a moderate CEC (21.38 $cmol_c$ $kg^{-1}$), likely due to its total OC content (1.43%) (Table 1).

**Table 1** Basic properties of the soil used in this study (mean ±SD).

| Parameters | Mean |
| --- | --- |
| Clay (%) | $14.41^{\pm1.16}$ |
| Silt (%) | $10.67^{\pm2.31}$ |
| Sand (%) | $74.92^{\pm2}$ |
| $pH_W$ | $4.84^{\pm0.02}$ |
| $pH_{Ca}$ | $4.21^{\pm0.01}$ |
| EA ($cmol_c$ $kg^{-1}$) | $2.95^{\pm0.14}$ |
| CEC ($cmol_c$ $kg^{-1}$) | $21.21^{\pm1.62}$ |
| TOC (%) | $1.43^{\pm0.04}$ |
| Total $Al^{3+}$ (g $kg^{-1}$) | $3.42^{\pm0.44}$ |
| Total $Fe^{2+}$ (g $kg^{-1}$) | $6.78^{\pm0.25}$ |

SD: standard deviation, $pH_W$: soil pH in deionised water, $pH_{Ca}$: soil pH in 0.01 M $CaCl_2$ solution, EA: exchangeable acidity, CEC: cation
115 exchange capacity, TOC: total organic carbon.

The mean $pH_W$ of the OAs varied between 5.50 (wheat straw) and 9.75 (biochar) (Table 2). EC of the amendments ranged from 0.55 dS $m^{-1}$ for faba bean straw to 3.22 dS $m^{-1}$ for compost. Excess cation content, which represents the acid neutralising capacity of the OAs and is expressed as the difference between total concentrations of cations and anions, was highest in compost (383 cmol $kg^{-1}$) and lowest in wheat straw (53 cmol $kg^{-1}$) (Table 2).

120 **Table 2** Chemical properties of organic amendments used in this study. Different letters indicate significant differences (p ≤0.05).

| Organic amendment | $pH_W$ (1:10) | EC (dS $m^{-1}$) | Na | K | Ca | Mg | P | S | Excess cations ($cmol_c$ | TOC (g $kg^{-1}$) | TN (g $kg^{-1}$) | C:N |
| --- | --- | --- | --- | --- | --- | --- | --- | --- | --- | --- | --- | --- |
| | | | | | cmol $kg^{-1}$ | | | g $kg^{-1}$ | | | | |



|  | (1:10) |  |  |  |  |  | kg⁻¹) |  |  |  |  |  |
|---|---|---|---|---|---|---|---|---|---|---|---|---|
| Wheat straw | 5.50[a] | 1.87[a] | 15.62[ad] | 28.89[a] | 11.28[a] | 6.68[a] | 0.27[a] | 1.02[a] | 53[a] | 453[a] | 7.30[a] | 63.15[a] |
| Faba bean straw | 6.96[b] | 0.55[b] | 3.12[b] | 7.17[b] | 69.46[b] | 13.52[ac] | 1.78[b] | 1.68[b] | 66[a] | 429[b] | 12.88[b] | 33.40[b] |
| Blended poultry litter | 7.88[c] | 1.39[c] | 12.20[a] | 15.49[c] | 168.95[c] | 41.33[b] | 2.47[c] | 4.32[c] | 187[b] | 256[c] | 17.09[c] | 15.09[cd] |
| Biochar | 9.75[d] | 0.91[d] | 25.03[c] | 15.74[c] | 108.82[d] | 22.16[c] | 1.11[d] | 1.12[a] | 154[b] | 180[d] | 9.63[a] | 18.96[c] |
| Compost | 8.12[e] | 3.22[e] | 17.24[d] | 29.94[a] | 87.67[e] | 327.15[d] | 4.27[e] | 5.94[d] | 383[c] | 160[d] | 20.18[c] | 7.95[d] |

EC: electrical conductivity, TOC: total organic carbon, TN: total nitrogen, C:N: carbon to nitrogen ratio

**2.4 Determination of lime requirement of acidic soil**

**2.4.1 Titration with calcium hydroxide**

Titrations and pH measurements in this study were carried out in 1:5 soil to deionised water or 0.01 M CaCl₂ solution ratios. First, 5 g of dry soil was weighed to 50 mL polyethylene tubes in triplicate and then 25 mL of deionised water or 25 mL of 0.01 M CaCl₂ solution were added into the tubes. The initial soil pH was measured after 30 minutes stirring and calibration of the pH meter with standard pH 7.0 and 4.0 buffers. Then, 0.5 mL aliquots of 0.022 M Ca(OH)₂ were added to the suspensions, continuously stirred, and pH measurements were taken at the end of each time interval. Based on Liu et al.

(2004), the reaction time allowed between consecutive titrations to obtain constant pH measurements was 30 minutes. The titrations with incremental additions of 0.022 M Ca(OH)₂ were carried out while recording the cumulative volume of the 0.022 M Ca(OH)₂ added versus the corresponding soil pH. A digital titrator (Burette Digital Titrette® Bottle-top Burette 50 mL Capacity Class A, Australian Scientific, Australia) was used for the titrations.

This data was used to plot a regression curve between incremental rates of equivalent CaCO₃ (Mg ha⁻¹) consumed and the

corresponding soil pH. The regression curve was used to derive the slope, which was then used to calculate the 30-minute pHBC (pHBC₃₀) of the acidic soil. The amount of base needed to neutralise H⁺ (independent variable) was on the x-axis and the change in pH (dependent variable) was on the y-axis to generate the slope of the titration curve as:

$$\text{Slope} = \frac{\Delta Y}{\Delta X} = \frac{\Delta pH}{V_{Ca(OH)_2}} = \frac{\Delta pH}{\Delta H^+} \tag{1}$$

where $\Delta Y$ is the change in pH, $\Delta X$ is the amount of base consumed or protons removed to neutralize acids, $V_{Ca(OH)_2}$ is the

volume of base consumed during titration (mL), $\Delta H^+$ is the mmol kg⁻¹ of H⁺ removed during titration. The amount of protons ($\Delta H^+$) neutralised during the titration is equivalent to the amount of Ca(OH)₂ consumed. From this model, the LR of the acidic soil can be calculated as:

$$LR = \frac{\Delta pH}{\text{Slope}} = \Delta pH \times pHBC \tag{2}$$

Given that the molecular weight of CaCO₃ is 100 g per mol, the amount of Ca(OH)₂ consumed by the acidic soil expressed

as CaCO₃ equivalent (Mg ha⁻¹) was calculated as:



$$CaCO_3 \ (Mg \ ha^{-1}) = \frac{V \ x \ M \ x \ 100 \ g \ CaCO_3 \ mol^{-1} \ x \ 10^{-9} \ x \ Wt. \ (kg \ ha^{-1})}{S \ (kg)} \qquad (3)$$

Where V is the volume (mL) of 0.022 M $Ca(OH)_2$ consumed to raise the initial pH of the acidic soil to the target pH, M = 0.022 is the molarity of $Ca(OH)_2$ in mmol $mL^{-1}$, $10^{-9}$ is the conversion factor from $CaCO_3$ (mg $mmol^{-1}$) to $CaCO_3$ (Mg $mmol^{-1}$), S is the mass of soil used for the titration (0.005 kg), and Wt is the weight of soil per hectare which in this case was

calculated as 1.5 x $10^6$ kg, assuming a liming depth of 0.1 m and soil bulk density of 1500 kg $m^{-3}$. By using these known variables in eqn.3, $CaCO_3$ (Mg $ha^{-1}$) equivalent can be simply calculated as V x 0.66.

The pH buffer capacity ($pHBC_{30}$), expressed in mmol $H^+$ (kg soil)$^{-1}$ $pH^{-1}$, was calculated from the titration curve as the inverse of the slope of the linear regression between pH and the added base (Mg $CaCO_3$ $ha^{-1}$) (Shi et al., 2019; Thompson et al., 2010). The unit of pHBC derived from the slope was Mg $CaCO_3$ $ha^{-1}$ $pH^{-1}$. This unit was converted to mg $CaCO_3$ $kg^{-1}$

$pH^{-1}$ and then to mmol $H^+$ $kg^{-1}$ $pH^{-1}$.

### 2.4.2 Equilibration experiment

An additional experiment was conducted to assess whether the 30-minute titration time above was sufficient to complete the equilibrium exchange reaction between $H^+$ and 0.022 M $Ca(OH)_2$. Since there was no significant difference between the slopes of regression lines fitted for titrations in water and 0.01 M $CaCl_2$ (p value of 0.231 at $\alpha = 0.05$), the equilibration

experiment was carried out only in a 1:5 soil to deionised water ratio. The experiment consisted of unamended soil (control) and soils amended with the equivalent titration point (ETP) multiplied by 0.5, 0.75, 1, 1.25, and 1.5. The ETP is the final volume of 0.022 M $Ca(OH)_2$ solution consumed over a 30-minute complete titration to achieve the target $pH_W$ of 6.5. These treatments were added in 50 mL polypropylene tubes with replicates of 5 g dried soil and 25 mL deionised water suspension. After stirring the mixture for 30 minutes, the initial 30-minute soil pH was measured, which was used to calculate the lime

buffer capacity ($LBC_{30}$) for each rate of 0.022 M $Ca(OH)_2$. Then, three drops of chloroform were added to minimise microbial activity. The tubes were covered with parafilm with only small opening left for air exchange to reduce evaporation, stirred regularly, and incubated at room temperature. The pH was measured every 24 h for 5 days while stirring the suspensions. The lime buffer capacity (mg $CaCO_3$ $kg^{-1}$ $pH^{-1}$) was calculated for each incubation period (Kissel et al., 2007):

$$LBC = \frac{V \ x \ M \ x \ M_W}{S \ x \ (pH_e - pH_o)} \qquad (4)$$

Where V is the volume of $Ca(OH)_2$ added (mL), M is the molarity of $Ca(OH)_2$ used for the titration (= 0.022 M), $M_W$ is the molecular weight of $CaCO_3$ (100 mg $mmol^{-1}$), S is the weight of soil titrated (kg), $pH_e$ is the pH of the suspension taken after addition of certain volume of $Ca(OH)_2$ and equilibration for specific incubation time, and $pH_o$ is the pH of a suspension without $Ca(OH)_2$ taken at similar incubation time.



Total acidity expressed as proton ($H^+$) concentrations that was neutralised over the incubation periods to finally attain equilibrium pH was calculated using eqn. 5, which was derived from $\Delta$pH x pHBC, by substituting 1/slope as pHBC in eqn.1.

$$H^+ = (LBC \text{ x } \Delta pH \text{ x } 2)/100 \tag{5}$$

Where $H^+$ is the proton concentration (mmol $H^+$ kg$^{-1}$ soil) that was neutralised at a given incubation time ($t_i$), LBC is the lime
buffer capacity (mg CaCO$_3$ kg$^{-1}$ pH$^{-1}$) calculated for the incubation time ($t_i$), $\Delta$pH is the difference between initial soil pH taken after 30-minutes of shaking the soil with base and the final soil pH taken at the incubation time ($t_i$), 100 is the conversion factor from mg CaCO$_3$ kg$^{-1}$ pH$^{-1}$ to mmol CaCO$_3$ kg$^{-1}$ pH$^{-1}$, and 2 is the conversion factor from mmol CaCO$_3$ to mmol $H^+$.

### 2.4.3 Equilibrium buffer curves and lime rates

Nonlinear regression curves between soil pH and incubation time as well as between LBC and incubation time were plotted for all concentrations of 0.022 M Ca(OH)$_2$ added to evaluate the change in soil pH and LBC over time. The curves were used to determine the equilibrium pH and LBC at equilibrium (LBC$_{eq}$, i.e. point where the pH did not change significantly with time). In addition, a linear regression was fitted between LBC$_{30}$ and LBC$_{eq}$, with different rates of 0.022 M Ca(OH)$_2$ as covariate. The regression equation between LBC$_{30}$ and LBC$_{eq}$ was used to calculate equilibrium lime buffer capacity (LBC$_{eq}$)
for the acidic soil based on the pHBC determined during the 30-minute titration, LBC$_{30}$. After the determination of LBC$_{eq}$, the lime requirement (LR) of the acidic soil was calculated as:

$$LR \text{ (mg CaCO}_3 \text{ kg}^{-1} \text{ soil)} = (LBC_{eq} \text{ x } (pH_t - pH_i))/ENV \tag{6}$$

Where LBC$_{eq}$ is the lime buffer capacity at equilibrium soil pH (mg CaCO$_3$ kg$^{-1}$ pH$^{-1}$), pH$_i$ is the initial pH$_W$ before addition of Ca(OH)$_2$, pH$_t$ is the target pH$_W$ (i.e. 6.5 used in this study), and ENV is the effective neutralising value (ENV) of the lime
used in percent.

### 2.5 Titratable alkalinity and application rate of organic amendments

The titratable alkalinity was determined by extracting the dried OAs with acid and then back titrating the suspension to pH 7 with base using the modified methods of Singh et al. (2017) and Yuan and Xu (2011). Briefly, 0.5 g of dried and sieved (0.25-2 mm) OAs were weighed into 50 mL polyethylene tubes in replicates. Then, 10 mL of 1 M HCl was added, and the
mixture was shaken on a reciprocal shaker for 2 h. The suspension was allowed to stand for 24 h and then titrated with 0.5 M NaOH using a digital titrator. The amount of acid neutralized by each amendment, expressed as mmol $H^+$ per gram OA, was calculated as the difference between the volume of 0.5 M NaOH consumed by the blank and the sample. The lime equivalent value, expressed as CaCO$_3$ equivalents, of the OAs was then calculated as:



$$LEV_{OA} = \frac{(0.5)*(b-s)*M_w}{2*W*1000}$$ (7)

Where LEV is the lime equivalent value (g CaCO₃ kg⁻¹ OA), 0.5 is the molarity of NaOH used for titration (mole L⁻¹), b is
the volume of NaOH consumed by the blank (mL), s is the volume of NaOH consumed by the sample (mL), $M_w$ is the
molecular weight of CaCO₃ (g mole⁻¹), 2 is the moles of H⁺ neutralised by one mole of CaCO₃, W is the weight of sample
(kg), and 1000 is the unit conversion factor. Then, the amounts of OAs required to raise soil pH to a desired level (in this
study pH$_W$ 6.5) were calculated by:

$RAR_{OA}$ (g OA kg⁻¹ soil) = $(LBC_{eq} * (pH_t - pH_i) * 1000)/LEV_{OA}$ (8)

Where $RAR_{OA}$ is the recommended application rate of OAs, $LBC_{eq}$ is the lime buffer capacity at equilibrium pH (mg CaCO₃
kg⁻¹ soil pH⁻¹), $pH_t$ is the target soil pH, $pH_i$ is the initial soil pH, and 1000 is the unit conversion factor.

### 2.6 Calculating lime and organic amendment combinations

    The application rates in the combinations of pure lime and OAs were calculated based on the titratable alkalinity of the OAs
and the CaCO₃ required to raise soil pH$_W$ to 6.5 that depends on $LBC_{eq}$. The method used to calculate lime-OA combinations
is illustrated in Fig. 1.

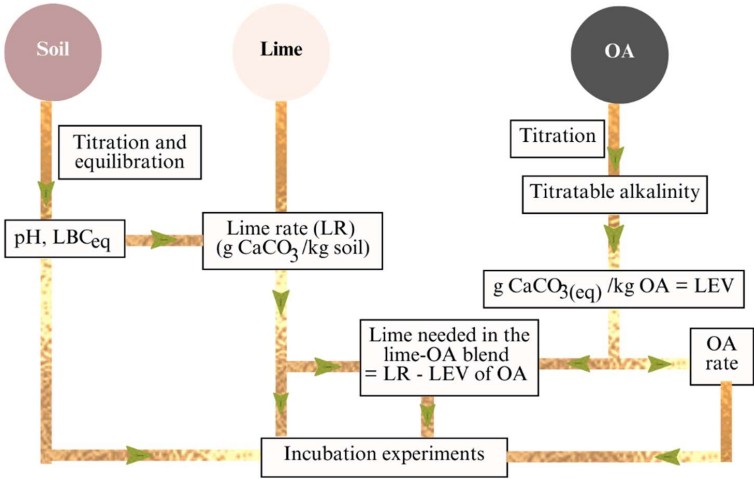

**Figure 1.** Conceptual method used to calculate the amount of lime, organic amendments (OA), and lime-OA combinations. $LBC_{eq}$:
equilibrium lime buffer capacity, LR: lime rate required to achieve pH$_W$ 6.5, LEV: lime equivalent value.



The amount of pure lime that was combined with each OA to achieve a target $pH_W$ of 6.5 was calculated by subtracting the LEV or $CaCO_3$ equivalent of the OA (i.e. added at 1.5% w/w in this study) from the lime rate using eqn. 9.

$$LOA_{comb} \ (g \ CaCO_3 \ kg^{-1} \ soil) = LEV_{OA} + \ LR_{inorg} \tag{9}$$

The $LOA_{comb}$ is the total $CaCO_3$ required in the lime-OA combinations to raise $pH_W$ to 6.5, $LEV_{OA}$ is the lime equivalent value or $CaCO_3$ content of each OA (i.e. added at 15 g OA $kg^{-1}$ in this study), and $LR_{inorg}$ is the amount of lime (i.e. inorganic

source of $CaCO_3$) needed in the combinations (g $CaCO_3$ $kg^{-1}$ soil).

### 2.7 Calculating organic amendments mixes

A 50:50% alkalinity-based mixture of selected OAs was prepared by combining rapidly decomposable organic material (wheat straw, faba bean straw) with resistant organic materials (compost, biochar) to test the effect of the mixture on soil pH over time. The amount of each amendment in the two OAs mix was calculated in order to contribute 50% of the total

alkalinity (i.e. 50% of 2.81 g $CaCO_3$ $kg^{-1}$) required to neutralise soil acidity and achieve $pH_W$ 6.5.

### 2.8 Validation experiments at different moisture contents

Incubation experiments were conducted at 60, 100, and 150% field capacity (FC) to validate whether the amounts of soil amendments calculated based on their titratable alkalinity could raise soil pH to the desired level ($pH_W$ 6.5). The treatments included unamended or control soil, lime, OAs, OAs mix, and lime-OA combinations. The amendment rates calculated

based on the method developed in this study are presented in Supplementary Table ST1.

For the incubations at 60 and 100% FC, the amendments were mixed with 50 g soil (< 2 mm) in pots in replicates. The pots were incubated in a dark room with deionized water added on a weight basis. The pH was measured after 14, 30, 60, 90, and 120 days for incubation at 60% FC and after 10, 20, 30, 40, and 50 days for the 100% FC until equilibrium pH was attained. For incubation at 150% FC, 5 g air-dry soil (< 2 mm) was weighed into 50 mL polyethylene plastic tubes and the

amendments were mixed with the soil.

The treatments were incubated in a dark room at 150% field capacity with moisture content maintained on a weight basis. Soil pH was measured at 4, 6, 8, 10, and 12 days by adding the amount of deionized water needed to make a 1:5 soil to water ratio and shaking the suspension for 45-minutes.

### 2.9 Statistical analysis

Regression curves for the titration points and equilibration experiments were fitted using OriginPro Lab version 2022 (9.95). Mean comparisons of titratable alkalinity and other chemical properties of OAs were carried out using One-Way ANOVA. Mean comparisons of pH values measured over time during titration and equilibration experiments, as well as the rate of $H^+$ neutralisation with incubation time, were undertaken using One-Way repeated measures ANOVA. Changes in pH values of





amended soils over time for validation experiments conducted at different soil water contents were also analysed using One-
Way repeated measures ANOVA. The mean comparisons, correlation analysis, and tests of significance were conducted at p
$\leq 0.05$ using IBM-SPSS version 28.0.1.0.

## 3 Results

### 3.1 Soil pHBC$_{30}$ and LBC$_{30}$

The regression lines between soil pH and incremental base additions (expressed as Mg CaCO$_3$ ha$^{-1}$ equivalents, eqn. 3) were
linear for titrations in water (r$^2$ = 0.996) and 0.01 M CaCl$_2$ (r$^2$ = 0.999) (Fig. 2). The slopes of the regression lines were 0.92
and 0.91 pH ha Mg$^{-1}$ CaCO$_3$ for titrations in water and 0.01 M CaCl$_2$ solution, respectively with no significant difference
between them (p = 0.231).

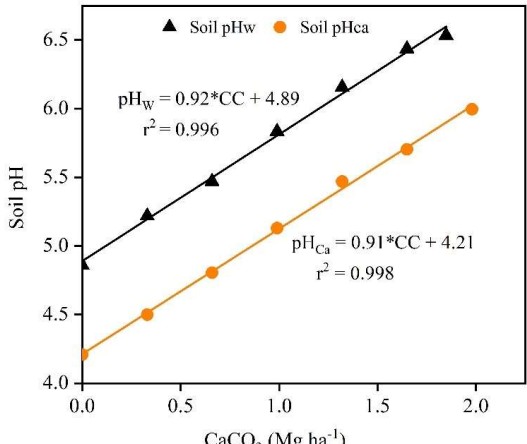

**Figure 2.** Fitted linear regression lines for titrations to pH 6.5 in deionised water and pH 6.0 in 0.01 M CaCl$_2$ solution after 30 minutes. In
the equations, 'CC' is the CaCO$_3$ (Mg ha$^{-1}$) consumed during titrations of the acidic soil with 0.022 M Ca(OH)$_2$.

The buffering capacity (pHBC$_{30}$) of the acidic soil was estimated from the inverse slope of the titration curve and was
1.09 Mg CaCO$_3$ ha$^{-1}$ pH$^{-1}$ in water and 1.10 Mg CaCO$_3$ ha$^{-1}$ pH$^{-1}$ in 0.01 M CaCl$_2$, which are equivalent to 14.54 and 14.66
mmol H$^+$ kg$^{-1}$ pH$^{-1}$, respectively. The pHBC is the amount of H$^+$ that is consumed to raise soil pH by one unit, whereas LBC
is the amount of CaCO$_3$ required to raise soil pH by one unit. As a result, pHBC expressed as LBC is more convenient for
calculating the amount of lime required to neutralise soil acidity. The average values of LBC$_{30}$ calculated from pHBC$_{30}$ were
727 mg CaCO$_3$ (kg soil)$^{-1}$ pH$^{-1}$ for titration in water and 733 mg CaCO$_3$ (kg soil)$^{-1}$ pH$^{-1}$ for titration in 0.01 M CaCl$_2$. The
slightly higher pHBC$_{30}$ in the 0.01 M CaCl$_2$ solution than in water indicates that soil resistance to pH change increases when





$Ca^{2+}$ ions in the $CaCl_2$ solution replace $Al^{3+}$ and $H^+$ ions in the soil exchange complex. The pHBC varied slightly across the pH values, indicating that the change in pH due to base additions is not uniform across the range of pH values.

270  **3.2 Equilibrium pH and lime buffer capacity**

The $pH_W$ and LBC of soils amended with 0.5, 0.75, 1, 1,25, and 1.5 times the ETP of 0.022 M $Ca(OH)_2$ and incubated for 5 days in a 1:5 soil to deionised water ratio are shown in Figs. 3a and 3b. The $pH_W$ of unamended soil and soils amended with different rates of 0.022 M $Ca(OH)_2$ decreased nonlinearly with incubation time until it reached a relatively constant level. In contrast, compared to its initial pH, $pH_W$ of the unamended soil increased slightly by 0.11 units by 72 h (Fig. 3a). To account

275  for this change, which was not associated with base addition, the pH value for lime-amended treatments was corrected by subtracting the change in pH of the control soil from the observed pH at the respective incubation time.

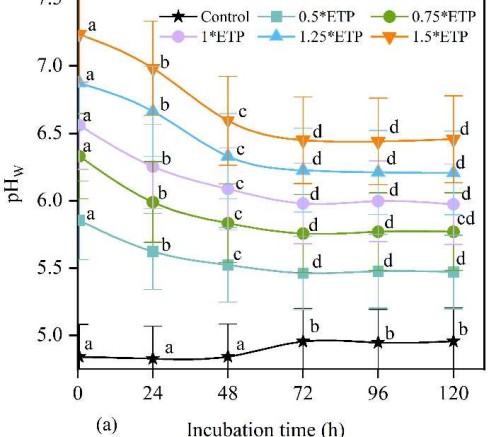

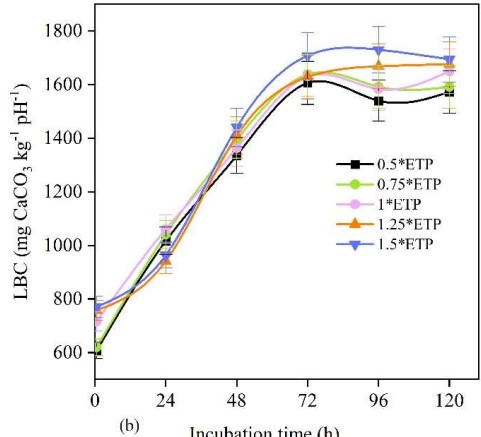



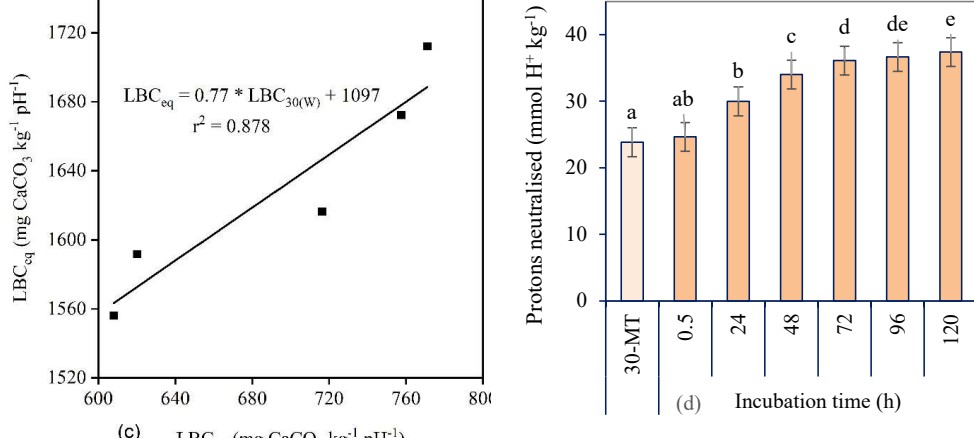

**Figure 3.** (a) $pH_W$ of unamended (control) and amended soils with incubation time, (b) relationship between lime buffer capacity (LBC –
mg $CaCO_3$ $kg^{-1}$ $pH^{-1}$) and incubation time. The LBC was calculated for different rates of 0.022 M $Ca(OH)_2$ using eqn. 4. The amended
soils received 0.022 M $Ca(OH)_2$ at rates of 0.5, 0.75, 1, 1.25, and 1.5*ETP where equivalent titration point (ETP) is the volume of 0.022 M
$Ca(OH)_2$ consumed during a 30-minute titration in water to raise soil $pH_W$ to 6.5 (i.e. 2.8 mL). (c) relationship between corrected $LBC_{eq}$
and $LBC_{30}$ of soils amended with different rates of 0.022 M $Ca(OH)_2$ converted to $CaCO_3$ equivalent. $LBC_{eq}$: equilibrium lime buffer
capacity (i.e. after 96-120 h), $LBC_{30}$: lime buffer capacity determined after 30-minute titration in water. (d) proton concentration
neutralised by 0.022 M $Ca(OH)_2$ added at a rate of 1*ETP during a 30-minute titration (30-MT) and 5-day incubation in water. Pairwise
comparisons using repeated measures ANOVA, different letters indicate significant differences (p ≤0.05) over time.

The soil $pH_W$ taken after 0.5 h for soil treated with 1*ETP was similar to those of the 30-minute titration. However, $pH_W$
values at 0.5, 24, and 48 h incubation were significantly different from each other and the rest of the incubation periods for
all rates of $Ca(OH)_2$. In contrast, the $pH_W$ values obtained after 72, 96, and 120 h incubation were similar. This indicates that
the acid-base reaction had reached equilibrium after 72 h incubation. Based on this and the trends of regression lines shown
in Fig. 3a, as well as the quantity of protons neutralised over time (Fig. 3d), the equilibrium pH was calculated as the average
of the adjusted pH values obtained after 96 and 120 h incubation. For example, the equilibrium pH for 1 ETP was 5.99 (i.e.
6+5.97)/2), implying that the 0.022 M $Ca(OH)_2$ equivalent determined during titration in water with 30 minute equilibration
time only neutralised 69% of the soil acidity when the target $pH_W$ was 6.5.

Furthermore, LBC was calculated using eqn.4 with incubation time adjusted with respect to the control (unamended) soil,
i.e., dividing the observed LBC by the correction factor (CF). The CF (eqn.10) was calculated by dividing the difference in
pH between a soil amended with '$a$' rate of 0.022 M $Ca(OH)_2$ and the control soil '$c$' at a specific incubation time '$t_i$' (e.g.,
24, 48, 72 h) by the difference in pH between a soil amended with the same rate of $Ca(OH)_2$ and the control measured at 0.5
h incubation as:





$$CF = (pH_a^{t_i} - pH_C^{t_i})/(pH_a^{0.5h} - pH_c^{0.5h}) \tag{10}$$

The LBC had an increasing non-linear trend with incubation time until equilibrium was reached, and values calculated for 72, 96, and 120 h incubations were not significantly different from each other (Fig. 3b) since LBC$_{eq}$ is obtained when the pH reaches equilibrium. By comparing the LBC values and the trends of regression lines in Fig. 3b, the average LBC values obtained at 96 and 120 h incubations were used as LBC$_{eq}$ for the acidic soil used in this study.

Subsequently, a regression equation was developed between adjusted LBC$_{eq}$ and LBC$_{30}$ (measured after 0.5 h), with different amounts of Ca(OH)$_2$ as covariate (Fig. 3c). The LBC$_{eq}$ for the soils incubated with base in deionised water was then calculated from LBC$_{30}$ value as:

$$LBC_{eq(W)} = 0.77 \text{ x } LBC_{30(W)} + 1097 \tag{11}$$

Where LBC$_{eq(W)}$ is the LBC$_{eq}$ for the soil incubated with 0.022 M Ca(OH)$_2$ and LBC$_{30(W)}$ is the LBC of soil for a 30-minute titration in deionised water. Using eqn. 10 and LBC$_{30}$ 727 mg CaCO$_3$ kg$^{-1}$ pH$^{-1}$, the LBC$_{eq}$ was 1657 mg CaCO$_3$ kg$^{-1}$ pH$^{-1}$. This relationship between LBC$_{30}$ and LBC$_{eq}$ can be used to calculate LBC$_{eq}$ from LBC$_{30}$ data for similar soils.

### 3.3 Proton concentrations and lime requirements

The cumulative concentrations of protons neutralised by the added base during the 5 days incubation with 1*ETP are shown in Fig. 3d. Thus, 1*ETP incubated for 120 h neutralised 37.39 mmol H$^+$ kg$^{-1}$ soil (eqn. 5) and raised pH$_W$ from 4.84 to 5.99, thus neutralising 36% more protons than the initial 30-minute titration (30-MT) (Fig. 3d). To raise the soil pH$_W$ to 6.5, 55 mmol H$^+$ kg$^{-1}$ must be neutralised. This corresponds to the total active acidity because base (OH$^-$) neutralises not only indigenous H$^+$ but also H$^+$ produced by hydrolysis of Al$^{3+}$. Therefore, 2751 mg CaCO$_3$ kg$^{-1}$ soil were needed to raise pH$_W$ to 6.5. For the lime used in this study (98% effective neutralising value) would be 2807 mg CaCO$_3$ kg$^{-1}$ soil (eqn.6).

### 3.4 Effect of the alkalinity of organic amendments on soil pH

The titratable alkalinity of OAs ranged from 12 cmol H$^+$ kg$^{-1}$ for wheat straw to 357 cmol H$^+$ kg$^{-1}$ for compost (Table 3). Titratable alkalinity of blended poultry litter and biochar was about half of that of compost (Table 3). The titratable alkalinity of OAs was most closely correlated with excess cation concentrations (r$^2$ = 0.98**, p <0.001), followed by EC of the amendments (r$^2$ = 0.69**, p = 0.004) (Supplementary Table ST2). In addition, alkalinity was moderately correlated with the inherent pH of OAs (r$^2$ = 0.60*, p = 0.018). The LEV or CaCO$_3$ content of each OA added at 1.5% (w/w) (i.e. 15 g kg$^{-1}$ soil) (eqn.7) varied with titratable alkalinity of the OA (Table 3). The amounts of OAs required to neutralise soil acidity and achieve pH$_W$ of 6.5, which was calculated based on LEV of the amendments and LBC$_{eq}$ of soil (eqn.8), was inversely related to the alkalinity content of the amendments (Table 3).




**Table 3** Titratable alkalinity of organic amendments and the amount of lime combined with the organic amendments added at 1.5% to raise soil $pH_W$ from 4.84 to 6.5. Different letters indicate significant differences (p ≤0.05).

| Type of organic amendment | Titratable alkalinity (cmol $H^+_{eq}$ kg$^{-1}$ OA) | LEV$_{OA}$ (g CaCO$_3$ kg$^{-1}$) | Calculated rates of OA (g kg$^{-1}$ soil) to achieve $pH_W$ 6.5 | LEV$_{OA}$ (g CaCO$_3$ (15 g OA)$^{-1}$) | Amount of lime (g CaCO$_3$ kg$^{-1}$ soil) required with 1.5% OA to achieve $pH_W$ 6.5 |
|---|---|---|---|---|---|
| Wheat straw | 11.7[a] | 5.8 | 471.1 | 0.09[a] | 2.72[a] |
| Faba bean straw | 43.0[b] | 21.5 | 127.8 | 0.32[b] | 2.49[b] |
| Blended poultry litter | 176.0[c] | 88.1 | 31.2 | 1.32[c] | 1.49[c] |
| Biochar | 168.7[c] | 84.4 | 32.6 | 1.27[c] | 1.54[c] |
| Compost | 357.0[d] | 178.7 | 15.4 | 2.68[d] | 0.13[d] |

OA: organic amendment, LEV: lime equivalent value. The sum of CaCO$_3$ of OA (column 5) and amount of lime (column 6) is equal to 100% CaCO$_3$ equivalent (2.81 g CaCO$_3$ kg$^{-1}$ soil for the soil used in this study).

Changes in soil pH caused by all OAs were significantly different at all water contents. The change in soil pH (ΔpH) by the OAs varied with soil water content (60, 100, and 150% FC). Incubation with faba bean and wheat straw at 60% FC increased $pH_W$ of the soil from 4.84 to 6.53 and 6.42, respectively (Fig. 4a). For the less biodegradable or resistant OAs

(ROAs) (i.e. blended poultry litter, compost, and biochar), the soil pH at 60% FC was lower than for crop straws (Fig. 4a). Biochar resulted in higher soil $pH_W$ (6.27) than the other ROAs.



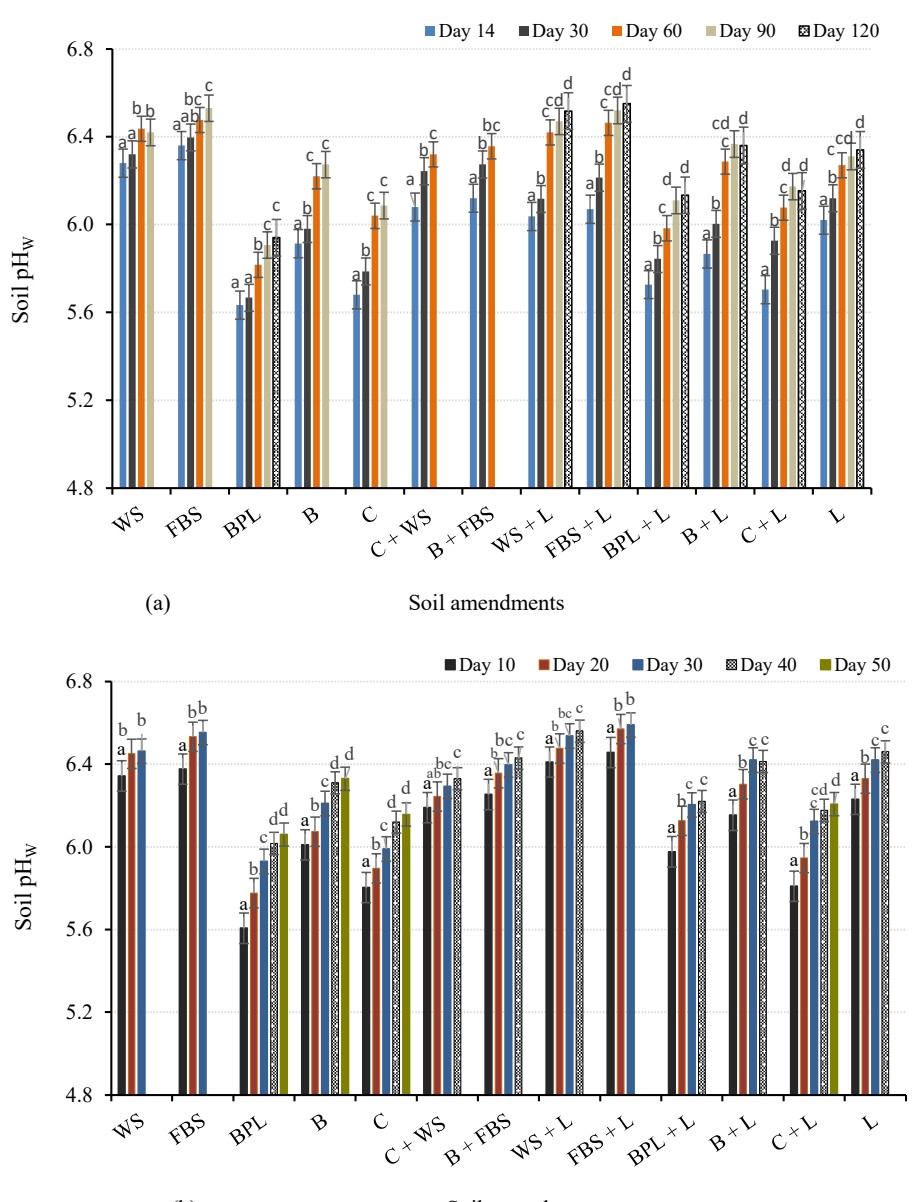

(a)     Soil amendments

(b)     Soil amendments



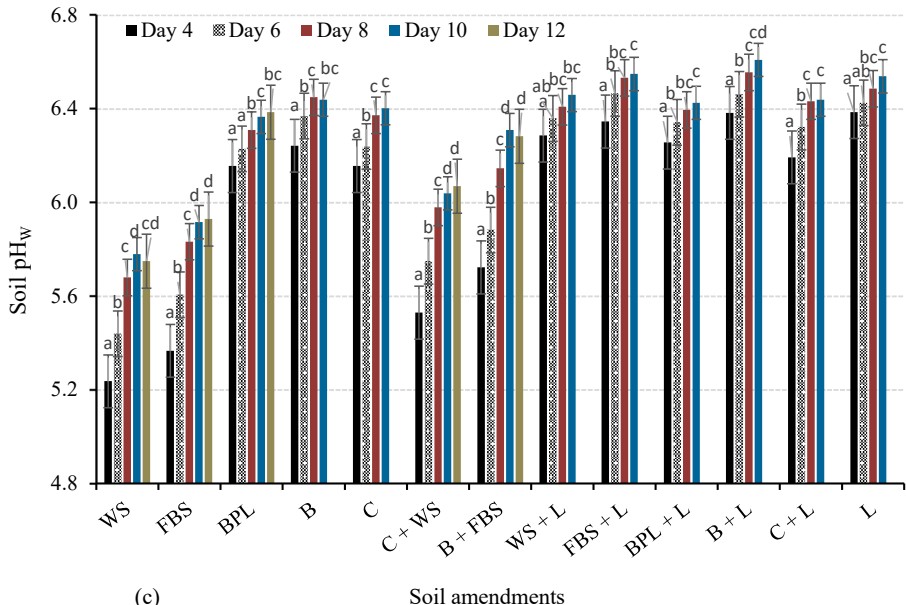

**Figure 4.** Changes in pHw of soils incubated with individual or combinations of different amendments (a) at 60% FC for 120 days, (b) at 100% FC for 50 days, and (c) at 150% FC for 12 days. Different letters indicate significant changes in soil pH with incubation time. The final pH values represent the equilibrium pH of the treatment. WS: wheat straw, FBS: faba bean straw, BPL: blended poultry litter, B: biochar, C: compost, L: lime.

At 100% FC, faba bean and wheat straws also resulted in soil pH higher than ROAs, with mean $pH_W$ values of 6.55 and
 6.46, respectively (Fig. 4b). At this soil moisture content, ROAs such as biochar and compost increased soil $pH_W$ from 4.84 to 6.33 and 6.16, respectively, thus higher pH values than at 60% FC (Fig. 4a). The rate of change in soil pH over time ($\Delta$pH/time in days) was also rapid and higher than at 60% FC. For instance, the rates of pH change for faba bean straw and biochar-amended soils were 0.22 and 0.13 pH units day$^{-1}$ at 100% FC, whereas 0.11 and 0.07 pH units day$^{-1}$ at 60% FC, respectively. Faba bean straw-amended soil reached equilibrium pH after 30 days of incubation at 100% FC, but after 90
 days at 60% FC (Figs. 4a & 4b). The remaining soil amendments had a similar decrease in equilibrium period with increase in soil water content. This could be due to the increased availability of both water and air for biodegradation of OAs at 100% FC, resulting in higher alkalinity generation and neutralisation of acidic soil.

Soils incubated with ROAs for 12 days at 150% FC (submerged conditions) had a higher soil pH than with faba bean and wheat straws (Fig. 4c). Biochar, compost and blended poultry litter raised soil $pH_W$ to 6.44, 6.40, and 6.39, respectively. On



the other hand, wheat and faba bean straws resulted in lower pH values (5.75 and 5.93) at 150% FC than at the lower water contents. For all soil amendments, the rate of change in soil pH at 150% was greater than at 100 and 60% FC. For example, at 150% FC, faba bean straw and biochar-amended soils reached equilibrium pH after 12 and 10 days (Fig. 4c), and their rates of change in pH were 0.48 and 0.64 pH units day$^{-1}$, respectively.

The magnitude of change in pH or alkalinity production by straws decreased above 100% FC whereas that of lime and less biodegradable OAs linearly increased with soil water content. The correlation between titratable alkalinity of amendments and changes in soil pH increased significantly with soil water content with r$^2$ values of 0.72**, 0.75**, 0.82** at 60, 100, and 150% FC, respectively (Supplementary Table ST2).

The alkalinity-based mixture of wheat straw + compost or faba bean straw + biochar aimed to generate 50:50% total alkalinity was calculated as 235.6 g kg$^{-1}$ wheat straw + 7.7 g kg$^{-1}$compost or 63.9 g kg$^{-1}$ faba bean straw + 16.3 g kg$^{-1}$
biochar. The mixes of resistant and easily biodegradable OAs are based on their alkalinity to provide the required 100% CaCO$_3$ to neutralise soil acidity. The 50:50% alkalinity-based mixture of resistant and easily decomposable OAs produced soil pH values that were intermediate between the individual amendments. At 60% FC, the wheat straw and compost mixture resulted in a soil pH$_W$ of 6.32, which was 0.23 units higher than the pH changes caused by compost but 0.10 units lower than with wheat straw (Fig. 4a). The faba bean straw and biochar mixture increased pH$_W$ to 6.36, resulting in values that were
higher than biochar but lower than faba bean straw. A similar trend was observed at 100% FC, but with slightly higher equilibrium pH values of 6.33 (compost – wheat straw mix) and 6.43 (biochar – faba bean straw mix) (Fig. 4b). This shows that the pH changes by alkalinity-based OA mixes can be predicted from pH changes by individual OAs.

### 4.5 Effect of lime – organic amendment combinations

The amount of lime required in the lime-OA combinations was calculated using eqn.9. When the OAs were added at 15 g kg$^{-1}$
, the amount of lime needed to ameliorate acidic soils in combined lime-OAs was inversely related to the titratable alkalinity of the OAs. With increasing alkalinity of the OAs from 5.8 g CaCO$_3$ kg$^{-1}$ (wheat straw) to 178.7 g CaCO$_3$ kg$^{-1}$ (compost), the lime requirement in lime-OA combinations decreased from 2.72 (wheat straw) to 0.09 g CaCO$_3$ kg$^{-1}$ (compost) (Table 3, Fig. 5). Compost addition for example, reduced the lime requirement by 95% compared to lime alone. Wheat straw with very low alkalinity, on the other hand, reduced lime requirement by only 3% compared to lime alone.



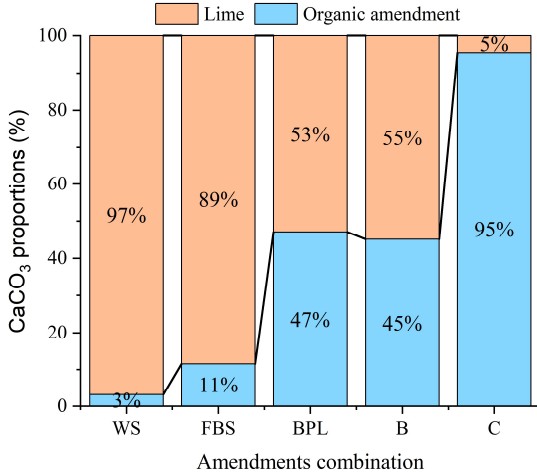


**Figure 5.** Proportions of alkalinity generated by organic amendments (OAs) added at 15 g kg$^{-1}$ and the lime required to achieve 100% CaCO$_3$ equivalent in combined lime-OA applications. WS: wheat straw, FBS: faba bean straw, BPL: blended poultry litter, B: biochar, C: compost.

Lime added alone increased soil pH by 0.20 units when soil moisture content increased from 60 to 150%. Combinations

of lime and OAs resulted in a relatively higher soil pH than OAs alone at all incubation moistures (Figs. 4a, 4b, & 4c). Combined additions of faba bean straw + lime resulted in soil pH$_W$ of 6.55 followed by wheat straw + lime (pH$_W$ of 6.52), after 120 days of incubation at 60% FC (Fig. 4a). At 100% FC, the faba bean + lime and wheat straw + lime combinations generated soil pH$_W$ 6.59 and 6.56, within 30 and 40 days of incubation, respectively (Fig. 4b). At 100% FC, lime combined with biochar, blended poultry litter, and compost increased soil pH$_W$ to 6.41, 6.22, and 6.21, respectively. At 150% FC, the

combined lime-ROA treatments increased soil pH$_W$ close to 6.5 (Fig. 4c).

## 4 Discussion

Our findings show that the laboratory method developed based on titratable alkalinity can be used to accurately calculate organic amendment rates and lime-organic amendment combinations for amelioration of acid soils.

### 4.1 Optimum application rates of soil amendments

The acidity in the soil that had to be neutralised to achieve the target of pH$_W$ 6.5 was estimated by determining an equilibrium LBC. In this study, an equilibrium pH of 5.99 was reached after 96-120 h at the equivalent CaCO$_3$ rate



determined during titration, giving an $LBC_{eq}$ of 1657 mg $CaCO_3$ $kg^{-1}$ $pH^{-1}$. Lime requirement was then calculated by multiplying this $LBC_{eq}$ by the required change in soil pH. However, it should be noted that the time required to reach equilibrium pH depends on the pHBC of the soil (Jalali and Moradi, 2020; Kissel et al., 2005; Wang et al., 2015). Stopping

incubations before equilibrium is reached may result in underestimation of lime requirement, whereas much longer incubations may lead to errors due to soil pH fluctuation caused by factors unrelated to the added base. Hence it is required to check this equilibrium time when using this method for different soils. LBC estimates based on 30-minute titration ($LBC_{30}$) did not represent the total concentrations of soil acidity that must be neutralised to bring the pH to the desired level, because the acid-base neutralisation reactions do not attain equilibrium in this short time period (Barrow and Cox, 1990;

Kissel et al., 2007; Liu et al., 2004). However, we showed a linear regression between $LBC_{eq}$ and $LBC_{30}$ can be used for rapid assessment, once this relationship is developed for a particular soil.

The application rates of OAs and lime-OA combinations required to neutralise soil acidity were calculated based on titratable alkalinity of OAs and the buffering capacity of the acidic soil. The results showed that the percent by weight (w/w) of OAs needed to achieve a soil $pH_W$ of 6.5 is inversely related to the titratable alkalinity of those amendments. The titratable

alkalinity of OAs increased in the order of wheat straw < faba bean straw < biochar < blended poultry litter < compost (Table 3), the optimum application rates of the amendments decreased as wheat straw (47.1%) > faba bean straw (12.8%) > biochar (3.3%) > blended poultry litter (3.1%) > compost (1.5%).

## 4.2 Effect of different organic amendment properties on soil pH

Alkalinity content has been previously found to be a primary measure of acid neutralizing capacity of OAs (Noble et al.,

1996; Noble and Randall, 1998). This was confirmed in the present study by the significant correlation between soil pH and titratable alkalinity of OAs at various soil water contents. The differences in alkalinity content of OAs has previously been found to be due to differences in excess cation content and other physicochemical properties which are influenced by soil, climate, and management practices (Noble and Randall, 1998; Slattery et al., 1991).

Previous studies have also showed that OAs generate alkalinity in the form of carbonates, organic anions (e.g. oxalate,

malate), and inorganic anions such as sulfate ($SO_4^{2-}$), phosphate ($PO_4^{3-}$), silicate ($SiO_4^{4-}$), and iron hydroxides ($FeO-O-$), which detoxify $H^+$, $Al^{3+}$, and other acid-forming ions through neutralisation and association reactions (Cai et al., 2020; Fidel et al., 2017; Li et al., 2022; Sakala et al., 2004). The hydroxyl ($-OH$) and carboxyl ($-COOH$) surface functional groups of organic materials also serve as the main proton acceptors in acidic soils (Dai et al., 2017; Tomczyk et al., 2020). Different OAs have different types and amounts of these functional components.

In our experiments the magnitude of soil pH changes at the same soil water content was not consistently related to the titratable alkalinity of the OAs added. This suggests other physicochemical properties of OAs (e.g. inherent pH, carbonate content, and C/N ratio) may significantly influence soil pH changes. Biochar, for example, was added at a rate of 100% $CaCO_{3eq}$, which was comparable to the rates for compost and blended poultry litter, but it increased soil pH more than these amendments. This might be caused by other properties of biochar such as high inherent pH and solid phase carbonate content



following pyrolysis (Mosley et al., 2015). Previous studies also found that significant increases in soil pH after amendment with biochar can be due to its unique properties, such as oxygen-rich surface functional groups and a large specific surface area, which leads to high surface adsorption of protons and $Al^{3+}$ (Chintala et al., 2014; Cui et al., 2021).

We found that when applied at similar $CaCO_3$ equivalents, faba bean straw produced higher soil pH than wheat straw at all soil water contents. This could be because faba bean straw has a lower C/N than wheat straw (Table 2) and therefore

decomposes faster, resulting in net ammonification during the mineralization of organic N and a pH increase. Higher total N concentrations in legumes than in cereals likely lead to a higher N cycling rates and a faster biological decarboxylation (Butterly et al., 2013), resulting in faster and greater soil pH increases.

**4.3 Impact of soil water content on alkalinity production by organic amendments**

The present study showed that changes in soil pH by wheat and faba bean straws decreased in the order 100 % FC > 60% FC

> 150% FC, whereas ROAs soil pH increased with water content as 60% FC < 100% FC < 150% FC. When applied at similar $CaCO_3$ equivalent, wheat and faba bean straws resulted in higher pH values than ROAs at 60% FC. The more rapid soil pH changes with wheat and faba bean straws to $pH_W$ 6.53 and 6.42 compared to the other OAs at 60% FC are likely because crop straws decompose fast because microbial activity is high when oxygen availability is high (Grzyb et al., 2021; Jin et al., 2023), leading to the rapid release of available alkalinity. In contrast, the pH changes with the ROAs were slow at

60% FC. The lignin content of wood chips in blended poultry litter and the resistance of compost and biochar to further decomposition may have resulted in slow decomposition of these amendments, thereby reducing the release of organic anions. In addition, the water content at 60% FC may not be high enough for the dissolution of soluble organic compounds and thus production of alkalinity in ROAs. As a result, soil amended with ROAs reached equilibrium pH after 90-120 days at 60% FC, indicating that ROAs should be added several months before planting to neutralise acidity.

We found that an increase in soil water content from 60% to 100% FC reduced the time required for amended soils to reach equilibrium pH nearly twofold. This suggests that soil water content plays an important role in the decomposition and dissolution of OAs to generate alkalinity. Thus, soil water content should be considered when deciding on the application time and comparing the effects of amendments on soil pH (e.g. optimal application may be before a significant rainfall event).

At 150% FC, amendment with ROAs resulted in $pH_W$ close to 6.5, likely because there was enough water for the dissolution of soluble organic/inorganic anions and carbonates. Even after the equilibrium pH was reached, undissolved/undecomposed ROA particles were found in the suspension, causing minor deviations in soil $pH_W$ from 6.5. However, it is unlikely that the entire acid-extractable alkalinity of the ROAs is released by water. Adeleke et al. (2017) suggested that organic anions that provide various functional groups (amino, carboxylic, phenolic) for surface adsorption of

$H^+$ and $Al^{3+}$ are weak acids and do not dissolve completely in water. Acid-extractable alkalinity determined by reaction of organic materials with acids (e.g. HCl) over a long time can also lead to increased solubility of inorganic alkali, exposing occluded inorganic alkali and occluded conjugate bases of functional groups (Fidel, 2012), which are not soluble in water.



Nevertheless, water-insoluble fractions of acid-extractable alkalinity can increase long-term buffering capacity of acidic soils (Fidel et al., 2017; Yuan et al., 2011).

However, wheat and faba bean straws induced lower pH at 150% FC. The lower pH with wheat and faba bean straws at 150% FC than at the lower water contents may be because undecomposed crop straws were added at high rates which resulted in low oxygen content at this high water content due to low decomposition rates and thus limited production of alkalinity. For example, Chen et al. (2018) showed that anaerobic conditions reduced the rate of straw decomposition by 30% compared with aerobic conditions. Reduction of organic material decomposition is associated with a limited production
of organic anions (Cai et al., 2020; Xu et al., 2006b). Furthermore, anaerobic conditions may have favoured the formation of acid-forming products such as $H_2S$, and promote the formation of protons from the acidic soil via the hydrolysis of $Al(OH)_3$ and the dissolution of Fe hydroxy oxide clay minerals.

### 4.4 Effect of mixing soil amendments based on alkalinity production

An alkalinity-based mixture of easily decomposable and ROAs produced soil pH changes intermediate between the two
amendments. The mixtures led to greater pH increases than ROAs alone, but smaller pH changes than rapidly decomposable alone. Previous studies indicated that the nutrient release and decomposition patterns of a mass-based OAs mixtures resulted in non-additive responses (Gartner and Cardon, 2004; Le and Marschner, 2018). However, there is no information on how alkalinity-based mixtures of OAs affect soil pH. Our findings show that for alkalinity-based OA mixtures, pH changes of the mixes can be predicted from the pH changes of the individual amendments. The pH values can be estimated as the sum of
pH changes by individual amendments multiplied by their proportion of alkalinity in the mix. Hence, mixing of OAs based on their alkalinity and C/N ratio could promote an early pH increase due to the rapid production of alkalinity by easily decomposable amendments, as well as a sustained increase later from the resistant amendments. Amendment mixtures may occur where crop residues are left in the field after harvest which are mixed with ROAs to generate the additional alkalinity required to neutralise soil acidity.

The results showed that lime-OA combinations induce higher pH than individual OAs. For instance, the combined lime-ROA treatments increased soil $pH_W$ to about 6.5 at 150% FC, which can likely be attributed to the increased alkalinity production for the reasons explained above. As lime dissolution in acidic soils increases with water content (Anderson et al., 2020; Naorem et al., 2022), combined use of lime and ROAs could be a good option for managing acidic soils forming in coastal areas and wetlands, including acid sulfate soils. Mixtures of lime and wheat and faba bean straws also produced
higher pH than lime alone. This could be due to synergetic interactions between lime and OAs. Decomposition of OAs increases $CO_2$ release due to increased microbial activity. The $CO_2$ reacts with water to form carbonic acid ($H_2CO_3$), which dissociates into $H^+$ and bicarbonates ($HCO_3^-$) (Bhatt et al., 2019). The $H^+$ ion then reacts with carbonates (e.g. $CaCO_3$) and converts them into bicarbonates (e.g. $Ca(HCO_3)_2$), which readily dissolve in water. This suggests that, depending on their decomposition rate, OAs can release $CO_2$ into soils, accelerating lime dissolution (Ahmad et al., 2014). Additionally, liming
may accelerate the decomposition of OAs due to lime-induced increases in microbial activity and soil respiration rates (Biasi



et al., 2008), leading to high alkalinity generation. However, the relative effectiveness of OAs with lime in ameliorating acidic soil is not simply the sum of their neutralising capacity (Butterly and Tang, 2018), likely due to the adsorption of cations (e.g. $Ca^{2+}$) released from lime on the surfaces of OAs.

**5 Conclusion**

Titratable alkalinity of OA and LBC of acidic soils can be used to accurately calculate the amounts of OAs and the lime-OA combination ratio needed to neutralise soil acidity. The alkalinity production potential of soil amendments is significantly affected by soil amendment type and soil water content. Crop straws decompose faster to release more alkalinity under aerobic than anaerobic conditions. On the other hand, lime and ROAs, such as compost, biochar, and blended poultry litter, lead to higher pH in soils with high water content. Alkalinity-based mixtures of two OAs produced soil pH changes that

were intermediate between the individual amendments. Lime-OA combinations lead to positive interactions, generating more alkalinity than individual amendments. We suggest that OAs with high alkalinity together with lime could reduce the amount of lime required and associated costs. This could be particularly important in situations where lime sources are unavailable or need to be transported long distances. The method could help to reduce the time and costs associated with conducting field experiments to determine the optimum lime-OA rates for a specific soil. It could also allow flexibility in

adjusting the amount of lime or OAs in blended applications based on their availability and cost. Uncertainty in achieving the desired soil pH from amendment calculations based on titratable alkalinity could be due to soil moisture following amendment, fractions of water-insoluble alkalinity, and technical errors in measuring soil $LBC_{eq}$ and OA alkalinity. Hence field validation of the method would be useful, particularly for the first time it is used for a particular soil or amendment.

**Data availability**

The data generated in this study are available from the corresponding author upon request.

**Author contributions**

BI, LMM, and PM conceptualized the study. BI conducted soil sampling and performed the experiment. BI drafted the manuscript with editorial contributions from both the co-authors.

**Competing interests**

The authors declare no competing interests.



**Acknowledgements**

Birhanu Iticha receives a University of Adelaide Research Scholarship and support from the GRDC project on management of subsoil acidity. We would also like to thank Brian Hughes, Ruby Zelda Hume, and Bonnie Armour for assistance in collecting the soil.

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
