# Peer review of "Combining lime and organic amendments based on titratable alkalinity for efficient amelioration of acidic soils"

_EGUsphere, 2023_

## Author Response (AR1)

**Authors' response to Referee 1 (Bernhard Wehr)**

General comments:

Reviewer: Addition of organic matter and lime to increase soil pH is a widely used practice and wood ash in particular has been used for centuries as a liming material. Hence the novelty and scientific significance of the work and its experimental approach is rather low.

Authors' response: The reviewer is right that lime and organic amendments have been used to ameliorate soil acidity for centuries. However, a laboratory method for calculating the lime and organic amendment (OA) combinations required to neutralise soil acidity is lacking. Currently agronomists use field experiments (which is expensive and time consuming) to determine the optimum lime-OA combinations. As a result, our new laboratory-based method for informing combinations of lime and OAs for amelioration of acid soils is novel for the following reasons:

- it helps to calculate lime-OA combinations required to ameliorate soil acidity without conducting field experiments and thus reduce the time and costs associated with field trials.
- It allows flexibility in adjusting the amount of lime or OAs in combined applications based on their availability and cost.
- It is reproducible in different soils as it only requires measurements of standard soil parameters (soil pH and lime buffer capacity, LBC) and titratable alkalinity of OAs.

Furthermore, recent research has shown that lime and OA combinations may be more effective than lime alone in neutralising soil acidity. Thus, this research is timely as it will greatly assist with calculating optimum lime and OA combinations.

Reviewer: While the methodology is well established and the execution of the experimental work appears good, there are some questions that need to be clarified. The presentation quality of the paper is overall good, but the number of formulae (some of which are not required to make their point) detract from the readability of the paper.

Authors' response: Two formulae were removed.

Specific comments:

Reviewer: The alkalinity of the organic amendments was estimated from the difference of sum of cations and sum of anions. This approach assumes that all anions and cations are accounted for. However, ICP will likely correctly quantify cations, but the sum of anions would be severely underestimated. The authors only report sulfate and phosphate as anions determined by ICP, but plant material would contain mainly nitrate and some chloride and bicarbonate rather than sulfate or phosphate, and certainly nitrate is not quanitified by routine ICP. The underestimation of anions results in an overestimation of the alkalinity of the organic amendments and incorrectly estimates (underestimates) the required organic amendment rate. The authors need to defend their rationale for their approach to estimating the alkalinity.

Authors' response: We did not consider excess cations when calculating OA or combined lime-OA application rates. To calculate soil amendment rates, we used the titratable alkalinity of OAs as well as standard soil parameters such as pH and $pHBC_{eq}$. Titratable alkalinity was determined by extracting the dried OAs with acid and then back titrating the suspension to pH 7 with base.

For clarity, we have removed the excess cations column from Table 2.

Reviewer: Since the authors have shown that the pH reaches an equilibrium after 72 hours (Figure 3b), it is not clear to me why the experiment work was conducted using 30 min equilibration times. The Dunn titration used to estimate the lime requirement of soils uses a 4 day (96 h) equilibration time. The authors should better explain why a 30 min equilibration time was used instead, since an incomplete equilibration will incorrectly estimate the required lime rate.

Authors' response: We did not calculate the lime rate using 30-minute equilibration (incomplete equilibration). The 30 minute equilibration was used to determine the equivalent titration point (ETP), which refers to the amount of base required to bring the initial pH of the acidic soil to $pH_w$ 6.5. We then performed a 5-day complete equilibration, which was used to calculate $LBC_{eq}$ and lime rate. Different soils incubated with base reach equilibrium pH at different times, depending on their buffering capacity of acidic soils. The LBC obtained in the 30-minute titration was subsequently used to develop a regression equation with $LBC_{eq}$. The relationship between $LBC_{30}$ and $LBC_{eq}$ can then be used to calculate $LBC_{eq}$ from $LBC_{30}$ data for

similar soils (see Fig. 3c), to save time without the need to conduct a long-term incubation each time.

Reviewer: Finally, the titratable acidity was estimated after incubating the organic amendments with 1 M HCl for 24 hours (section 2.5). Did the authors check that this does not introduce artifacts by either increasing functional groups (e.g. demethylation of pectin) or degrading functional groups?

Authors' response: We do not think reaction of organic materials with 1M HCl for 24 hours will introduce significant artefacts with regard to determining titratable acidity. Strong acid pre-treatment is a routine procedure for analysis of organic carbon content in soils. Silveira et al. (2008) found that this method reliably estimated C pools in soils but some labile organic compounds (i.e. carbohydrates and amino acids) were removed by HCl treatment and washing with deionised water; however in our case these labile compounds remain in the titration solution as there is no washing step. This solution is then back-titrated so the acid-neutralising effects of any labile compounds mobilised are still measured. Furthermore, several studies have been published on titratable alkalinity of OAs using this acid concentration. It is considered a standard and reliable parameter used to calculate the acid-neutralizing value of OAs. Our results showed that OA rates calculated using titratable alkalinity produced soil pH values close to 6.5, especially for rapidly decomposable organic materials. However, as shown in this study, other factors such as soil water content also influence pH change.

Reference: Silveira et al. (2008). Characterization of soil organic carbon pools by acid hydrolysis. *Geoderma* 144, 405-414.

Reviewer: Adding up these concerns, I have severe reservations about the validity of the study in its present form.

Authors' response: Our responses above address the concerns raised and Reviewer 2 did not share similar concerns.

Technical corrections:

L25. State that this value refers to pHw.
Authors' response: Thank you. Done.

L83-84. Rephrase/reword
Authors' response: Thank you. Revised.

**Authors' response to Anonymous Referee #2**

The paper aimed how lime and organic amendments could affect soil pH and if it was possible to predict such increase using the amendment alkalinity. The manuscript is overall well-written.

Therefore, I recommend its publication after few minor revisions.

Authors' response: Thank you for your very positive comments.

**Abstract and Graphical Abstract**

Clear and informative

Authors' response: Thank you.

**Introduction**

Line 51: the ")" after "unit" should be removed.

Authors' response: Done.

The introduction is well-written and gives an overview of the subject and the objectives clearly.

Authors' response: Thank you.

**Materials and Methods**

Lines 110-112: this is not part of the collection. This entire section is actually giving the results of the previous section (analysis of soil and amendments).

Authors' response: Thank you for the comment. Subheading 2.3 is "Properties of soil and amendments", but not "collection of soil and amendments". Now it has been revised.

Table 2: why is there no replicate in this case?

Authors' response: All the treatments had three replications. This is now clarified in the table caption by adding (n=3).

The methods used, and calculations are well described.

Authors' response: Thank you.

**Results**

Line 293: one bracket is missing.

Authors' response: The bracket was added.

Table 3: since statistical analysis was performed, replicates must have been done. Thus, the authors should give the SD of the measurements.

Authors' response: SD has been added in the revised version.

The Results are clearly described but a bit long.

Authors' response: The reviewer is right. We carried out many experiments to address the title and objectives. As a result, the results were a bit long.

**Discussion**

The discussion is well developed.

Authors' response: Thank you.

**Conclusion**

The conclusion is supported by the results and well-written.

Authors' response: Thank you.

---

## Author Response (AR2)

**Response to the reviewer comments**

We thank the reviewer for his comments. Below, our responses to the reviewer comments are in red font.

Thank you to the authors for addressing points I raised in the earlier version of the paper. I still have a few issues with the paper.

Firstly, the data presented are only valid for this one soil, the results are not applicable to other soils.

Authors' response: This is incorrect. The manuscript describes a method developed for calculating OA application rates and lime-OA combinations based on measurements of standard soil parameters (pH, buffering capacity) and titratable alkalinity of OAs. As the above-mentioned soil parameters are measured using standard methods (e.g. see Aitken and Moody 1994, Kissell et al. 2019), this method is applicable to other soils. This was already mentioned in the previous revised manuscript and in our earlier response to the reviewer. Soil amendment calculations for any soil can be done using this method. The method was tested on five different types of OAs incubated in sandy loam soil.

References:

Aitken, R. L., and Moody, P. W.: The effect of valence and ionic-strength on the measurement of pH buffer capacity. Soil Research, 32(5), 975-984, https://doi.org/10.1071/SR9940975, 1994.

Kissel, D., Sonon, L., and Cabrera, M.: Rapid measurement of soil pH buffering capacity. Soil Science Society of America Journal, 76(2), 694-699, https://doi.org/10.2136/sssaj2011.0091, 2012.

Secondly, reporting buffer capacity as delta pH is misleading; for instance, raising pH 5 to 5.7 (delta pH 0.7) is not the same as raising pH 6 to 6.7: there is tenfold difference in actual protons, despite delta pH being the same. I wonder if it would not better to convert pH to actual protons concentrations and base the paper on changes in protons rather than changes in pH. This will also allow to do away with correction factors, and may help address issues raised by the authors (L 425-426; L 296-303; L344-356)?

Authors' response:

We did not report buffer capacity as delta pH, buffer capacity was calculated by dividing delta H$^+$ by delta pH. Specifically, as stated in the method section of the manuscript "The pH buffer capacity (pHBC$_{30}$), expressed in mmol H$^+$ (kg soil)$^{-1}$ pH$^{-1}$, was calculated from the titration curve as the inverse of the slope of the linear regression between pH and the added base". As such the delta pH following base additions is just used to normalise the pH change and is relative to different initial soil pH values. This is a standard method and

nomenclature for expressing pH buffer capacity (e.g. Aitken and Moody 1994, Kissell et al. 2019). We also used delta pH to represent a change in pH of amended soil due to proton neutralisation by the amendments.

Finally, the Discussion is too long and could be condensed quite a bit.

Authors' response: The discussion was shortened by about 20%.

**Editor's comments**

Please, revise the manuscript again according to the new comments of the reviewer, particularly those regarding the excessive length of the discussion section.

Authors' response: The discussion was shortened by about 20%.